# A Stretchable, Transparent, and Mechanically Robust Silver Nanowire–Polydimethylsiloxane Electrode for Electrochromic Devices

**DOI:** 10.3390/polym15122640

**Published:** 2023-06-10

**Authors:** Tingting Hao, Leipeng Zhang, Haoyu Ji, Qiyu Zhou, Ting Feng, Shanshan Song, Bo Wang, Dongqi Liu, Zichen Ren, Wenchao Liu, Yike Zhang, Jiawu Sun, Yao Li

**Affiliations:** 1School of Materials Science and Engineering, Harbin Institute of Technology, Harbin 150001, China; htt12030810@163.com; 2Center for Composite Materials and Structure, Harbin Institute of Technology, Harbin 150001, China; ziegqqzieg@gmail.com (Q.Z.); wb641368602@163.com (B.W.); zyk302838165@163.com (Y.Z.); 3Infrared and Low Temperature Plasma Key Laboratory of Anhui Province, NUDT, Hefei 230037, China; 4School of Chemistry and Chemical Engineering, Harbin Institute of Technology, Harbin 150001, China; yu000204@163.com (H.J.); songshanshan0502@126.com (S.S.); ldq416abc@163.com (D.L.); 19s018113@stu.hit.edu.cn (Z.R.); 22b325005@stu.hit.edu.cn (W.L.); 17390616271@163.com (J.S.); 5School of Materials Science and Engineering, Qingdao University of Science and Technology, Qingdao 266042, China; fengting@qust.edu.cn

**Keywords:** silver nanowire, electrochromic devices, polydimethylsiloxane, flexible smart windows, Tungsten trioxide

## Abstract

The application of flexible indium tin oxide (ITO-free) electrochromic devices has steadily attracted widespread attention in wearable devices. Recently, silver nanowire/poly(dimethylsiloxane) (AgNW/PDMS)-based stretchable conductive films have raised great interest as ITO-free substrate for flexible electrochromic devices. However, it is still difficult to achieve high transparency with low resistance due to the weak binding force between AgNW and PDMS with low surface energy because of the possibility of detaching and sliding occurring at the interface. Herein, we propose a method to pattern the pre-cured PDMS (PT-PDMS) by stainless steel film as a template through constructed micron grooves and embedded structure, to prepare a stretchable AgNW/PT-PDMS electrode with high transparency and high conductivity. The stretchable AgNW/PT-PDMS electrode can be stretched (5000 cycles), twisted, and surface friction (3M tape for 500 cycles) without significant loss of conductivity (ΔR/R ≈ 16% and 27%). In addition, with the increase of stretch (stretching to 10–80%), the AgNW/PT-PDMS electrode transmittance increased, and the conductivity increased at first and then decreased. It is possible that the AgNWs in the micron grooves are spread during PDMS stretching, resulting in a larger spreading area and higher transmittance of the AgNWs film; at the same time, the nanowires between the grooves come into contact, thus increasing conductivity. An electrochromic electrode constructed with the stretchable AgNW/PT-PDMS exhibited excellent electrochromic behavior (transmittance contrast from ~61% to ~57%) even after 10,000 bending cycles or 500 stretching cycles, indicating high stability and mechanical robustness. Notably, this method of preparing transparent stretch electrodes based on patterned PDMS provides a promising solution for developing electronic devices with unique structures and high performance.

## 1. Introduction

In recent years, the development of flexible transparent conductive electrodes (FTE) has become the focus of various electronic technology applications, such as electronic paper, flexible displays, variable batteries, and wearable sensors [1]. Indium tin oxide (ITO) and fluorinated tin oxide (FTO) are still the market’s most commonly used conductive materials [2]. However, their drawbacks of brittleness and high cost restrict their further application in the field of flexibility. Therefore, several novel alternative materials have been developed to replace ITO and FTO-based transparent electrodes, including graphene [3], conductive polymers [4], carbon nanotubes [5], metal grids [6], metal nanowires [2,7,8], and hybrid electrodes [9,10,11]. Silver nanowires (AgNWs) are considered the most promising candidate due to their excellent optoelectronic properties and outstanding mechanical flexibility.

At the same time, with the rapid development of wearable electronics, the requirement for intrinsically stretchable devices becomes extraordinarily urgent. Stretchable electronic devices are becoming a promising new technology for the next generation of wearable devices because they can adapt to multi-angle deformation and provide a comfortable sensation for human skin. As the core component of stretchable electronic devices, the preparation and use of stretchable electrodes have received considerable attention. Polydimethylsiloxane (PDMS) is one of the most promising stretchable flexible substrate materials owing to its excellent chemical and thermal stability, strong elasticity, and high transmittance [2,12]. Although the research on AgNW/PDMS-based FTE has achieved tremendous progress in many flexible electronic fields, it is still challenging to maintain electrode performance stability (detaching and sliding) under long-term bending or even tensile conditions. Due to the large difference in elastic modulus between the AgNWs conductive network and PDMS substrate, the bending or stretching process often produces deformation at the contact interface. As a result, the nanowires were separated from the PDMS substrate, causing the shedding of nanowires or electrode failure.

To address the above challenges, the wrinkled structure formed by pre-strain and the semi-embedded structure formed by pre-curing are the common solutions for the preparation of the tensile electrode. Recently, Wallace et al. [13] prepared wrinkled AgNW films by releasing the pre-strained composite electrode, which has been demonstrated to be a promising approach to improving tensile stability. However, due to the low surface energy of PDMS or other elastomers, this wrinkled structure cannot solve the adhesion between the AgNW and the substrate. Wang et al. [14] prepared an ultra-thin PDMS-immobilized layer, constructed a semi-embedded wrinkled AgNW network, and overcame the adhesion problem by spin-coating pre-polymerized PDMS solution. However, in the process of spin coating, the thickness of pre-polymerized PDMS is difficult to control. Too thick a PDMS will reduce the conductivity of the electrode, and too thin will affect the adhesion of the AgNW network to PDMS. Lin et al. [12] developed a method that combines vacuum filtration with screen printing to fabricate AgNWs/PDMS electrodes. However, this method does not seem suitable for large-scale commercial preparation of electrodes. Therefore, the development of a scalable and facile preparation method to prepare AgNW-based flexible scalable electrodes without loss of mechanical properties or conductivity is still the core problem to be solved.

In this paper, a simple and effective PDMS grid-patterning method is presented to obtain AgNW-based FTE with excellent mechanical stability. The pre-cured PDMS surface was gridded and reprinted by the stainless steel film template. Then, the PT-PDMS with micron grooves on the surface was prepared. AgNWs are embedded into the groove by rod coating to form an AgNWs/PT-PDMS electrode. AgNWs were embedded into the groove by rod coating to prepare AgNWs/PT-PDMS electrodes. The gridded conductive network prevents the shedding and sliding of AgNWs during the process of friction; at the same time, the connection between grooves makes the electrode maintain high conductivity in the stretching process. Compared with other electrode preparation methods, the primary differences and advantages of this approach are as follows: (1) The gridded conductive network improves the mechanical stability of AgNW. (2) The problem of adhesion between AgNW and PDMS is solved in a simple way. As a result, the AgNWs/PT-PDMS electrode is highly robust in repeated bending 10,000 cycles or 500 peeling-off cycles in a 3M tape test without significant failure in conductivity. In addition, we demonstrated that the electrochromic electrodes based on AgNWs/PT-PDMS had been constructed, which exhibit noticeable electrochromic performance (transmittance contrast from ~61% to ~57%) and stretching properties. The proposed method for the surface gridding of PDMS is a promising strategy for the fabrication of flexible, stretchable electronic devices.

## 2. Experimental Section

### 2.1. Reagents and Materials

PDMS (SYLGARD 184) was purchased from Dow Corning. Analytical pure silver nitrate (AgNO_3_), analytical pure sodium chloride (NaCl·2H_2_O), and polyvinylpyrrolidone (PVP, K60: Mw #60000) powder were purchased from Aladdin. Ethanol and ethylene glycol were purchased from Xilong Technology & Chemical Co., Ltd., Shantou, China. Stainless steel film (hole diameter 60 μm) cleaned with acetone and ethanol was purchased from Changzhou Ningcong screen Co., Ltd., Changzhou, China. All chemicals were used as received without further purification.

### 2.2. Synthesis of AgNWs

Firstly, 30 mL ethylene glycol (EG) solutions of polyvinylpyrrolidone (2.0 g, PVP) and silver nitrate (0.51 g, AgNO_3_) were prepared. Then, the AgNO_3_ solution was added dropwise into the PVP solution for the initial nucleation of the silver seeds. Next, sodium chloride (0.6 mL, NaCl, 0.032 mol/L) solution was added to the mixture. The solutions were then solvothermal treated at 170 °C for 3 h. After the reaction, the solid was washed with deionized water and ethanol to remove EG, PVP, and other impurities from the supernatant. AgNWs dispersed in ethanol with a concentration of 2 mg/mL.

### 2.3. Preparation of PT-PDMS Film

The stainless steel film transfer printing method prepared PT-PDMS film with micro-porous grooves. The PDMS precursor was obtained by mixing the base and curing agent of Sylgard 184 (Dow Corning) with a ratio of 10:1, and the liquid mixture was degassed. The precursors were pre-polymerized at 60 °C for 30 min to form pre-cured PDMS (Pre-PDMS). The prepared Pre-PDMS films were cut into strips with sizes of 10 cm × 5 cm. Next, stainless steel films with the same area were attached to the surface of Pre-PDMS and polymerized at 60 °C for 10 h. Then, the composite film (SF-PDMS) was immersed in hydrochloric acid to remove the stainless steel film on the surface, and the PT-PDMS film with micro-porous grooves was obtained.

### 2.4. Preparation of AgNW/PT-PDMS and WO_3_/AgNW/PT-PDMS Electrode

The schematic diagram of the fabrication procedure for the AgNW/PT-PDMS electrode is depicted in Figure 1. In a typical synthesis procedure, the AgNWs solution with a concentration of 2 mg/mL was uniformly coated on the PT-PDMS film with micro-grooves and a Meyer rod. Then, the AgNW/PT-PDMS PDMS flexible electrode was obtained after drying in an environment with an oven temperature of 60 °C for 10 min. For comparison, the AgNW/PDMS electrode was also prepared by the same method, in which the flexible substrate was PDMS with a smooth surface.

Using electron beam evaporation technology, the active material WO_3_ was evaporated onto a AgNW/PT-PDMS substrate (thickness 300 nm, pressure 5 × 10^−4^ Pa) to obtain the WO_3_/AgNW/PT-PDMS electrode. Meanwhile, WO_3_/AgNW/PDMS electrodes were prepared for comparison.

### 2.5. Characterization

The crystalline phase and lattice parameter of the material were characterized by an X-ray diffraction spectrometer (XRD, D8 Bruker), where the XRD peaks ranged from 10° to 80° with a step size of 0.02°. The surface morphology and microstructure of the electrode were characterized by a field emission scanning electron microscope (SEM, Hitachi S-4800). The chemical composition of the electrodes was determined by XPS spectroscopy (XPS Thermo Fisher, E. Grinstead, UK). The film’s resistance is measured by a four-probe tester purchased by the company. The bending and tensile tests of the electrode were carried out with a self-made stretching machine, and the surface adhesion of the electrode was tested with 3M tape. The specific operation step of the self-made stretching machine is to bend or stretch the conductive electrode for a certain number of cycles and test the resistance change in the testing process to obtain the mechanical stability of the electrode. In the process of measuring the square resistance of the electrode, the resistance at five points was measured. In order to characterize the replicability of the results, an error bar is added to the results. The electrochemical performance of the electrodes was measured by a three-electrode system, in which the composite electrode was used as the working electrode, the Pt wire was used as the counter electrode, and the Ag wire was used as the reference electrode. Spectral measurements of the electrodes were recorded using a VIS-NIR fiber optic spectrometer (MAYA 2000-Pro, Ocean Optics, Tianjin, China).

## 3. Results and Discussion

### 3.1. Structure and Morphology

Figure 1 shows the method of constructing a AgNW/PT-PDMS flexible electrode with a micro-porous groove structure on the surface to obtain FTE with high mechanical properties. The stainless steel film is used as a sacrificial template and pasted onto the surface of Pre-PDMS to obtain SF-PDMS composite film. Then, the SF-PDMS film is etched with hydrochloric acid to remove the stainless steel film, and the micro-porous groove structure is formed on the PT-PDMS film. After this, AgNWs solution is coated in micro-porous grooves on the surface of PT-PDMS film by rod coating. In the process of bending and stretching, it is not easy to slip and peel off due to the fixation of AgNW in the micropore groove on the surface of PDMS, which improves the mechanical stability of the electrode.

To understand the formation of micron grooves and its effect on the distribution of AgNWs, the electrodes are characterized by SEM. Figure 1a,b show that the plasma-treated PDMS surface has a corrugated structure. Compared with the surface of PDMS, the surface of PT-PDMS film has an obvious microporous groove structure (see Figure 1c). Figure 1d,f are enlarged images of yellow marked areas in Figure 1c,e, respectively. As can be seen from Figure 1d, the average diameter of the groove is ~50 um. Then, to improve the mechanical flexibility of the electrode, the AgNWs solution is coated in the grooves on the PT-PDMS, as shown in Figure 1e,f. The AgNWs are spread evenly inside the grooves and are connected to form a network. The grooves and grooves on the surface of the PDMS were connected so that the circuit of the whole network was in the state. The AgNWs in the AgNW/PDMS film are uniformly spread on its surface, as shown in Appendix A. The morphology of the AgNWs prepared in the experiment is shown in Appendix A in the support information. The surface of AgNWs is smooth, and the length and diameter are about 60 μm and 50 nm, respectively. In order to better understand the preparation process of PT-PDMS substrate, the physical pictures were displayed. Figure 1g shows a photographic image of the stainless steel film with a dense pore diameter of about 60 μm. Figure 1h shows physical photos of PT-PDMS and PDMS substrates, marked with red dotted lines in the middle. Compared with the smooth PDMS surface, the PT-PDMS surface has obvious roughness. However, the surface roughness of PT-PDMS has little effect on the transmittance of the film (as shown in Figure 1i, the transmittance at wavelength 550 nm is ~90%).

Figure 2 shows how silver nanowires are embedded inside PDMS using rod coating and the state of AgNWs during peel testing. Firstly, the PT-PDMS with micro-porous structure is shown. The AgNWs solution was dripped onto the surface of PT-PDMS, and the solution was evenly coated with a Meyer rod. AgNWs are pushed into micropore grooves by Meyer rods and connected to each other to form a network. After repeated adhesion with 3M adhesive tape, the AgNW network structure will not be destroyed because it is fixed in the micropore slot on the surface of PDMS.

The structure of AgNW/PT-PDMS is further evaluated by the XRD. The diffraction patterns of the films are shown in Appendix A and indexed to face-centered cubic Ag (JPDCS No. 04-0783) [9]. The obvious four peaks at 38.11°, 44.30°, 64.44°, and 77.40° corresponded to (111), (200), (220), and (311) Bragg reflections of Ag. Thus, it is proven that the conductive electrode by AgNWs solution in the PDMS groove structure has been successfully developed.

To further evaluate the atomic species and bonding characteristics information of AgNW/PT-PDMS, the XPS and CHN analysis techniques were used. From the binding energy ranges of 0–550 eV, the typical Ag, C, O, and Si elements are detected through the full spectrum (Appendix A). For further determining the valence states of elements in AgNW/PT-PDMS, high-resolution analyses of Ag3d are carried out. Appendix A shows two peaks at 368.2 and 374.2 eV, belonging to signatures of Ag 3d5/2 and Ag 3d3/2 of metallic Ag, respectively [15]. The composition of the electrode was further analyzed by CHN analysis technology, as shown in the results of Appendix A. These results verify the formation of the AgNW/PT-PDMS nanocomposite structure.

### 3.2. The Optoelectrical Properties of Electrodes

The transmittance and electrical properties of the electrode are evaluated. Figure 2a shows that the optical transmittance of PDMS and PT-PDMS films at 550 nm wavelengths is 92% and 90%, respectively, indicating that the micro-porous groove structure does not affect the transmittance of the films. To balance the optical transmittance and conductivity of the electrode, the amount of AgNWs is optimized. The amount of AgNWs increased from 1 mL to 6 mL, and a series of conductive electrodes were prepared and evaluated. With the increase of AgNWs, the transmittance of AgNW/PDMS and AgNW/PT-PDMS electrodes decreased (Figure 2b). When the load of AgNWs is 6 mL, the transmittance of the AgNW/PDMS electrode at 550 nm is 37%, while that of the AgNW/PT-PDMS electrode is 63%. The reason may be that AgNWs are different in the spreading region of AgNW/PDMS and AgNW/PT-PDMS electrodes. The whole surface of the AgNW/PDMS electrode is coated with nanowires, while in the AgNW/PT-PDMS electrode, the nanowires are only spread in the grooves. The area of nanowires covered by AgNW/PDMS electrodes is larger than that of AgNW/PT-PDMS electrodes, which leads to a serious decrease in transmittance. The conductivity of the electrode increases with the increase of AgNWs, as shown in Figure 2c. When the amount of AgNWs is 1 mL, the conductivity of AgNW/PT-PDMS is better than that of the AgNW/PDMS electrode, which may be because there are more nanowires in the unit groove structure than on the PDMS surface. In order to balance the transmittance and conductivity of the electrode, we chose the amount of AgNWs solution of 4 mL to prepare the electrode.

To further evaluate the performance and advantages of AgNW/PT-PDMS electrodes in different harsh working conditions, the mechanical properties of AgNW/PT-PDMS electrodes are investigated. The mechanical properties are studied by a stretching machine, and the electrical conductivity changes of AgNW/PT-PDMS electrodes during bending and stretching were measured (Figure 3a,b). As shown in Figure 3a, after 5000 bending cycles with a radius of 1.0 cm, the sheet resistance growth rate of the AgNW/PDMS electrode increased by 200%. The reason for the increase in resistance may be that the surface of PDMS is very smooth, which causes AgNWs to fall off from the electrode surface after 5000 bending cycles, and the circuit on the surface of the AgNW/PDMS electrode is broken. In contrast, after 10,000 bending cycles, the resistance growth rate of AgNW/PT-PDMS electrodes remains stable, indicating that they have excellent bending durability under bending stress.

Apart from bending properties, stretchability is another important quality of flexible conductive electrodes. Figure 3b exhibits the cyclic stability of the electrode at 50% strain. As shown, the AgNW/PDMS electrode lost its conductivity after 5000 stretching cycles. Compared with AgNW/PDMS, AgNW/PT-PDMS electrodes can still maintain a relatively low resistance change after the stretching cycle, indicating their excellent stretchability (ΔR/R_0_ ≈ 16.0%). The reason may be that the micron groove structure on the surface of the AgNW/PT-PDMS electrode can not only inhibit the shedding of AgNWs from the surface but also enhance the stretching degree of the electrode. With the increase of tensile strain (tensile to 10–80%), the transmittance of the AgNW/PT-PDMS electrode increased, and the conductivity increased at first and then decreased, as shown in Figure 3c. The changing trend of electrical conductivity can also be seen from the changes in red triangles and black circles in Figure 3c. This may be due to the diffusion of AgNW in micron grooves during PDMS stretching, resulting in a larger unfolding area and higher transmittance of AgNW films. At the same time, the nanowires between the grooves touch each other, thus improving the electrical conductivity.

In addition, the effect of micron grooves on preventing AgNWs shedding on the PT-PDMS surface is evaluated by measuring the change of electrode conductivity after peeling with 3M tape. As shown in Figure 3d, the sheet resistance of the AgNW/PDMS electrode dramatically increased by 579% after 50 peeling cycles, indicating that the AgNWs network lost its conductive function. In contrast, the AgNW/PT-PDMS electrode remains relatively stable after 500 peel cycles (ΔR/R ≈ 27%). This may be because AgNWs are embedded in micron grooves to prevent stripping from the PT-PDMS film. In order to demonstrate the advantages of AgNW/PT-PDMS electrodes in commercial applications, AgNW/PT-PDMS is compared with previously reported electrodes, as shown in Appendix A. Compared with other commercial materials, we can observe that the mechanical stability of AgNW/PT-PDMS electrodes is more stable [16,17,18,19,20,21,22].

### 3.3. The Electrochromic Performance of Electrodes

Electrochromism is a phenomenon that the optical properties of electrochromic materials can be continuously converted reversibly under a certain external voltage [23]. The combination of electrochromic technology and flexible wearable technology is expected to produce novel electrochromic devices, which have great potential in smart clothing and implantable displays in the future. WO_3_ is the most widely studied cathode electrochromic oxide, which has long-term cycle stability and obvious color-switching characteristics. The studies show that the electrochromic behavior of WO_3_ is an ion and electron double intercalation/extraction model, which is caused by e^-^ and Li^+^ intercalation/extraction [24]. The studies show that the electrochromic behavior of WO_3_-based electrodes is due to the e^−^ and Li^+^ double intercalation/extraction process into WO_3_. The coloration (blue/charge) and bleaching (transparent/discharge) processes are based on the following reversible reactions [25]:(1)$${\;\mathrm{WO}}_{3}\left(\mathrm{bleached}\right)+{\mathrm{xe}}^{-}+{\mathrm{xLi}}^{+}\underset{\;}{\Leftrightarrow }{\mathrm{Li}}_{x}{\mathrm{WO}}_{3}\left(\mathrm{colored}\right)$$

Figure 4a shows the optical transmission spectra of the WO_3_/AgNW/PT-PDMS electrode at +1.0 V and −1.0 V voltage. The initial optical transmittance of the WO_3_/AgNW/PT-PDMS electrode at 550 nm is 73%. When a voltage of −1 V(vs. Ag/AgCl electrode) is applied to the electrode, the color of the electrode changes from transparent to dark blue with an optical transmittance of 10% at 550 nm. For comparison, we prepared WO_3_/AgNW/PDMS electrodes. To evaluate the cyclic stability of the WO_3_/AgNW/PT-PDMS electrode, the switching stability of the electrode during cyclic bleaching and coloration processes in the voltage range of +1.0 V to −1.0 V was investigated, which has great significance for WO_3_/AgNW/PT-PDMS electrode application in flexible electronic devices. The WO_3_/AgNW/PT-PDMS electrode exhibited switching stability with coloring and bleaching states. The colored state of the electrode is uniform, as shown in Figure 4b, and the coloring and bleaching states are reversible under different voltages. Figure 4c,d show the changes in transmittance of WO_3_/AgNW/PDMS and WO_3_/AgNW/PT-PDMS electrodes at 550 nm as a function of operating time. Each cycle consists of a positive bias for the 30 s and a negative bias for another 30 s. The results show that the ΔT (optical contrast between the coloration (−1.0 V) and bleached (+1.0 V) state) of the WO_3_/AgNW/PDMS electrode degraded after 10 cycles, from the initial 56% to 4%. The reason may be the poor adhesion between AgNWs and PDMS film surface, resulting in the AgNWs falling off from the PDMS surface, as shown in the illustration of Figure 4c. On the other hand, the WO_3_/AgNW/PT-PDMS electrode shows high optical contrast and good cyclic stability, and ΔT has almost no degradation (as shown in the illustration of Figure 4d). The results show that micron grooves can improve the bonding strength between AgNWs and PT-PDMS. Appendix A shows the bleaching and coloration processes of the WO_3_/AgNW/PT-PDMS electrode during 100 cycles. It can be found that the performance of the electrode degrades in the later stage of the cycle. The reason may be the oxidation of AgNW during the long-term electrochemical cycle, which leads to the degradation of the performance of the electrode.

To evaluate the mechanical properties of the electrode in different harsh working environments, the electrochromic stability of the electrode before and after repeated bending and stretching is investigated. Figure 5a shows the dynamics of the colored and bleached state of the electrode after bending 10,000 cycles at 550 nm. The ΔT of the WO_3_/AgNW/PDMS electrode decays seriously in the cycle after bending. Compared with WO_3_/AgNW/PDMS, the WO_3_/AgNW/PT-PDMS electrode remains stable (transmittance contrast from ~61% to ~57%), indicating that the electrode has excellent bending flexibility (Figure 5a). Stretchability is another important feature of WO_3_/AgNW/PT-PDMS electrodes. Figure 5b shows the coloring and bleaching cycle stability of the electrode before and after 500 stretches at 50% strain. After stretching for 500 cycles, the WO_3_/AgNW/PT-PDMS electrode can still keep the ΔT relatively stable, indicating its good extensibility. However, the ΔT of the WO_3_/AgNW/PDMS electrode degrades seriously after stretching, and the color state is almost unchanged. Therefore, the above results show that the WO_3_/AgNW/PT-PDMS electrode with microporous groove structure has excellent mechanical cycle stability.

## 4. Conclusions

In summary, we demonstrated AgNW/PT-PDMS electrodes with excellent optical and mechanical properties through a simple and scalable transfer method. The microporous groove structure is formed on the pre-cured PDMS surface by stainless steel film transfer printing, which improves the adhesion between AgNWs and PDMS. The AgNW/PT-PDMS electrode overcomes the disadvantages of AgNWs shedding and sliding on the surface of PDMS. The stretchable AgNW/PT-PDMS electrode can maintain stable mechanical properties after 10,000 bending cycles and stretching to 50%. The electrochromic electrode prepared by AgNW/PT-PDMS exhibits excellent flexibility and electrochromic behavior even after 500 stretching cycles. We believe that this novel structure of AgNW/PT-PDMS electrode has an important application prospect in the next generation of flexible electronic devices.

## Data Availability

Not applicable.

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
