# Peer review of "A Stretchable, Transparent, and Mechanically Robust Silver Nanowire–Polydimethylsiloxane Electrode for Electrochromic Devices"

_polymers, 2023, doi:10.3390/polym15122640_

Round 1
Reviewer 1 Report
An ITP free Flexble and transparent AgNW/PT-PDMS electrode as been reported by authors for application in electrochromic devices. The manuscript can be accepted only after addressing following comments:
1. The statement of innovation was not sufficient in the section of Introduction. Introduction should be clearly stated research questions and targets first. Then answer several questions: Why is the topic important (or why do you study on it)? What are research questions? What has been studied? What are your contributions? The innovative and main contributions should be discussed in a more detailed way in light of a broader related literature and in terms of generalization of its findings.
2. What is the state art of AgNW/PT-PDMS from the commercial point of view? Give a brief table about the comparison of AgNW/PT-PDMS with other commercial materials.
3. What is the predominant parameter in these influence factors? Construction of composite material, fabrication techniques, composition, morphology.
4. Chemical nature significantly control the physiochemical properties of coating film. So, authors must have to provide XPS and CHN analysis.
Minor editing of English language required
Author Response
Our responses to Reviewers’ comments are listed as follows:
Reply to Reviewer 1’s Comments:
Reviewer #1: An ITP free Flexible and transparent AgNW/PT-PDMS electrode as been reported by authors for application in electrochromic devices. The manuscript can be accepted only after addressing following comments:
1) The statement of innovation was not sufficient in the section of lntroduction. Introduction should be clearly stated research guestions and targets first, Then answer several questions: Why is the topic important (or why do you study on it)? What are research questions? What has been studied? What are your contributions? The innovative and main contributions should be discussed in a more detailed way in light of a broader related literature and in terms of generalization of its findings.
Response: Thanks for the reviewer’s valuable comment. According to the suggestion of reviewer, we have revised the contents of the introduction. In the introduction, the purpose and goal of the research are stated, and then the relevant contents are answered and the innovation and main contributions are discussed in more detail. Please see the yellow marking of the Introduction in the revised manuscript.
In recent years, the development of flexible transparent conductive electrodes (FTE) has become the focus of various electronic technology applications, such as electronic paper, flexible displays, variable batteries, and wearable sensors.1 Indium tin oxide (ITO) and fluorinated tin oxide (FTO) are still the market’s most commonly used conductive materials.2 However, its drawbacks of brittleness and high cost restrict its further application in the field of flexibility. Therefore, several novel alternative materials have been developed to replace ITO and FTO-based transparent electrodes, including graphene,3 conductive polymers,4 carbon nanotubes,5 metal grids,6 metal nanowires,2, 7-8 and hybrid electrodes.9-11 Silver nanowires (AgNWs) are considered to be the most promising candidate due to their excellent optoelectronic properties and outstanding mechanical flexibility.
At the same time, with the rapid development of wearable electronics, the requirement for intrinsically stretchable devices becomes extraordinarily urgent. Stretchable electronic devices are becoming a promising new technology for the next generation of wearable devices due to they can adapt to multi-angle deformation and provide a comfortable sensation for human skin. As the core component of stretchable electronic devices, the preparation and use of stretchable electrodes have received considerable attention. Polydimethylsiloxane (PDMS) is one of the most promising stretchable flexible substrate materials owing to its excellent chemical and thermal stability, strong elasticity, and high transmittance.2, 12 Although the research on AgNW/PDMS-based FTE has achieved tremendous progress in many flexible electronic fields, it is still challenging to maintain electrode performance stability (detaching and sliding) under long-term bending or even tensile conditions. Due to the large difference in elastic modulus between AgNWs conductive network and PDMS substrate, the bending or stretching process often produces deformation at the contact interface. As a result, the nanowires were separated from the PDMS substrate, causing the shedding of nanowires or electrode failure.
To address the above challenges, the wrinkled structure formed by pre-strain and the semi-embedded structure formed by pre-curing is the common solutions for the preparation of tensile electrode. Recently, Wallace et al.13 prepared wrinkled AgNW films by releasing the pre-strained composite electrode, which has been demonstrated to be a promising approach to improving tensile stability. However, due to the low surface energy of PDMS or other elastomers, this wrinkled structure cannot solve the adhesion between AgNW and substrate. Wang et al.14 prepared an ultra-thin PDMS-immobilized layer, constructed a semi-embedded wrinkled AgNW network, and overcame the adhesion problem by spin-coating pre-polymerized PDMS solution. However, in the process of spin coating, the thickness of prepolymerized PDMS is difficult to control. Too thick PDMS will reduce the conductivity of the electrode, and too thin will affect the adhesion of AgNW network to PDMS. Lin et al.15 developed a method that combines vacuum filtration with screen printing to fabricate AgNWs/PDMS electrodes. However, this method does not seem to be suitable for large-scale commercial preparation of electrodes. Therefore, the development of a scalable and facile preparation method to prepare AgNW-based flexible scalable electrodes without loss of mechanical properties or conductivity is still the core problem to be solved.
In this paper, a simple and effective PDMS grid-patterning method is presented, to obtain AgNW-based FTE with excellent mechanical stability. The pre-cured PDMS surface was gridded and reprinted by the stainless steel film template, then the PT-PDMS with micron grooves on the surface was prepared. AgNWs are embedded into the groove by rod coating to form an AgNWs/PT-PDMS electrode. AgNWs were embedded into the groove by rod coating to prepare AgNWs/PT-PDMS electrodes. The gridded conductive network prevents the shedding and sliding of AgNWs during the process of friction; at the same time, the connection between grooves makes the electrode maintain high conductivity in the stretching process. Compared with other electrode preparation methods, the primary differences and advantages of this approach are as follows: (1) the gridded conductive network improves the mechanical stability of AgNW; (2) the problem of adhesion between AgNW and PDMS is solved in a simple way. As a result, the AgNWs/PT-PDMS electrode is highly robust in repeated bending 10,000 cycles or 500 peeling-off cycles in a 3M tape test without significant failure in conductivity. In addition, we demonstrated the electrochromic electrodes based on AgNWs/PT-PDMS have been constructed, which exhibit noticeable electrochromic performance (transmittance contrast from ∼61% to ∼57%) and stretching properties. The proposed method for the surface gridding of PDMS is a promising strategy for the fabrication of flexible stretchable electronic devices.
- What is the state art of AgNW/PT-PDMS from the commercial point of view? Give a brief tableabout the comparison of AgNW/PT-PDMS with other commercial materials
Response: Thanks for the reviewer’s valuable comment. According to the suggestion of the reviewer, we have added the comparison of AgNW/PT-PDMS with other commercial materials. In order to demonstrate the advantages of AgNW/PT-PDMS electrodes in commercial applications, AgNW/PT-PDMS is compared with previously reported electrodes, as shown in table Table S1. Compared with other commercial materials, we can observe that the mechanical stability of AgNW/PT-PDMS electrodes is more stable. Please see the yellow marking in Table S1 in the revised supplementary.
Table S1 Optoelectronic properties and mechanical flexibility of Various Reported FTEs in comparison with AgNW/PT-PDMS
Samples |
Transmittance (%)/Sheet resistance (Ω/sq) |
Bending cycles |
Peeling off cycles |
Ref. |
AgNW/PEDOT:PSS/PET |
89.2/9.4 |
2000 |
30 |
1 |
MXene/AgNW-PVA |
52.3/18.3 |
1000 |
- |
2 |
AgNW-PVA |
87.5/63 |
250 |
- |
3 |
AgNW/PDMS |
75/20 |
1000 |
- |
4 |
AgNW/PET |
95/10 |
1000 |
- |
5 |
AgNW@TiO2-PI |
95/43.2 |
5000 |
100 |
6 |
AgNW/PT-PDMS |
74/6 |
10000 |
500 |
Our work |
- What is the predominant parameter in these influence factors? Construction of composite material, fabrication techniques, composition, morphology.
Response: Thanks for the reviewer’s valuable comment. According to the performance of AgNW/ PDMS-based electrode, the structure, preparation technology, composition and morphology of the composite will affect the performance of the electrode to a certain extent. At present, it is found that the core problem in the preparation of AgNW/ PDMS electrode is the unstable performance caused by the poor adhesion between AgNWs and PDMS substrate. Therefore, at present, the main influence technology is the suitable preparation process of AgNW/ PDMS electrode. In this paper, the surface of PDMS was gridded by a simple stainless steel template transfer printing method to obtain the electrode with excellent performance.
- Chemical nature significantly control the physiochemical properties of coating film. So, authors must have to provide XPS and CHN analysis
Response: Thanks for the reviewer’s valuable comment. According to the suggestion of the reviewer, we have added the XPS and CHN analysis. To further evaluate the atomic species and bonding characteristics information of AgNW/PT-PDMS, the XPS and CHN analysis techniques were used. From the binding energy ranges of 0–550 eV, the typical Ag, C, O and Si elements are detected through the full spectrum (Figure S3a). For further determining the valence states of elements in AgNW/PT-PDMS, high-resolution analyses of Ag3d are carried out. Figure S3b shows two peaks at 368.2 and 374.2 eV, belonging to signatures of Ag 3d5/2 and Ag 3d3/2 of metallic Ag respectively. The composition of the electrode was further analyzed by CHN analysis technology, as shown in the results of Table S2. These results verify the formation of AgNW/PT-PDMS nanocomposite structure.
Figure S3 XPS spectra of AgNW/PT-PDMS (a) and Ag 3d.
Table S2 CHN analysis of AgNW/PT-PDMS
Samples |
N(%) |
C(%) |
H(%) |
S(%) |
AgNW/PT-PDMS |
0.00 |
35.76 |
5.96 |
0.00 |
References
(1) Li, X.; Yu, S.; Zhao, L.; Wu, M.; Dong, H. Hybrid PEDOT:PSS to obtain high-performance Ag NW-based flexible transparent electrodes for transparent heaters. Journal of Materials Science: Materials in Electronics 2020, 31 (10), 8106-8115, DOI: 10.1007/s10854-020-03351-5.
(2) Zhou, B.; Su, M.; Yang, D.; Han, G.; Shen, C. Flexible MXene/Silver Nanowire-Based Transparent Conductive Film with Electromagnetic Interference Shielding and Electro-Photo-Thermal Performance. ACS Applied Materials & Interfaces 2020, XXXX (XXX).
(3) Zeng, X. Y.; Zhang, Q. K.; Yu, R. M.; Lu, C. Z. A new transparent conductor: silver nanowire film buried at the surface of a transparent polymer. Advanced Materials 2010, 22 (40), 4484-4488.
(4) Liu, H. S.; Pan, B. C.; Liou, G. S. Highly transparent AgNW/PDMS stretchable electrodes for elastomeric electrochromic devices. Nanoscale 2017, 9.
(5) Highly Efficient and Bendable Organic Solar Cells with Solution-Processed Silver Nanowire Electrodes. Advanced Functional Materials 2014, 23 (34), 4272-4272.
(6) Huang, Y.; Tian, Y.; Hang, C.; Liu, Y.; Wang, S.; Qi, M.; Zhang, H.; Peng, Q. TiO2-Coated Core-Shell Ag Nanowire Networks for Robust and Washable Flexible Transparent Electrodes. ACS Applied Nano Materials 2019.
(7) Kumaresan, Y.; Min, G.; Dahiya, A. S.; Ejaz, A.; Shakthivel, D.; Dahiya, R. Kirigami and Mogul-Patterned Ultra-Stretchable High-Performance ZnO Nanowires-Based Photodetector. Advanced Materials Technologies.
(8) Lin, Y.; Li, Q.; Ding, C.; Wang, J.; Yuan, W.; Liu, Z.; Su, W.; Cui, Z. High-resolution and large-size stretchable electrodes based on patterned silver nanowires composites. Nano Research 2022, 15 (5), 4590-4598, DOI: 10.1007/s12274-022-4088-x.

Reviewer 2 Report
This manuscript describes the properties of patterned PDMS embedded with AgNW for use as wearable devices.
Main points were to report on bendability, stretchability, and transmittance of device.
The manuscript is easy to read and well organized. Some of the background of polymers and carbon nanomaterials is mentioned in the introduction section.
Line 149 is repetitive to lines 146-148. Can be rewritten to be more concise.
Figure S1, a and b, would fit nicely as part of Figure 1 in main text. That would provide all surfaces being compared in this manuscript. a-f.
For section "The optoelectrical properties of electrodes" beginning in Line 168, was the optimal volume of AgNW volume explicitly stated in text?
As a comment, this section was the hardest to read. Lines 168-220.
How was the WO3 incorporated into the AgNW/PT-PDMS structure? Section beginning in Line 226.
Not mentioned are future work and optimization of other parameters to improve structure integrity.
In my opinion, there are too few citations (17 in total). Perhaps including growth of market or interests in this area of wearable devices with statistics or percentages would more robustly show the demand for this type of research.
Author Response
Reply to Reviewer 2’s Comments:
Reviewer #2: This manuscript describes the properties of patterned PDMS embedded with AgNW for use as wearable devices. The main points were to report on the bendability, stretchability, and transmittance of the device. The manuscript is easy to read and well-organized. Some of the backgrounds of polymers and carbon nanomaterials are mentioned in the introduction section.:
1) Line 149 is repetitive to lines 146-148. Can be rewritten to be more concise.
Response: Thanks for the reviewer’s valuable comment. According to the suggestion of reviewer, we have made corrections according to the reviewer's comments. “Compared with the surface of PDMS, the surface of PT-PDMS film has an obvious microporous groove structure (see Figure 1a). As can be seen from Figure 1b, the average diameter of the groove is ~50um. Then, to improve the mechanical flexibility of the electrode, AgNWs solution is coated in the grooves on the PT-PDMS as shown in Figure 1c and 1d.” Please see the yellow marking of the Introduction in the revised manuscript.
- Figure S1, a and b, would fit nicely as part of Figure 1 in the main text. That would provide all surfaces being compared in this manuscript. a-f.
Response: Thanks for the reviewer’s valuable comment. According to the suggestion of reviewer, we have revised Figure 1 and Figure S1. Please see the yellow marking in Figure 1 and Figure S1 in the revised manuscript and supplementary.
Figure 1. SEM images: AgNW/PDMS (a,b at different magnifications); PT-PDMS (c, d at different magnifications); AgNW/PT-PDMS (e, f at different magnifications). Physical pictures: Stainless steel film (g); PDMS and PT-PDMS (h); PT-PDMS (i).
- For the section "The optoelectrical properties of electrodes" beginning in Line 168, was the optimal volume of AgNW volume explicitly stated in text?
Response: Thanks for the reviewer’s valuable comment. According to the suggestion of reviewer, we have added a description of the optimal volume of AgNW volume selection. “In order to balance the transmittance and conductivity of the electrode, we chose the amount of AgNWs solution of 4ml to prepare the electrode.” Please see the yellow marking in Line 168 in the revised manuscript.
- As a comment, this section was the hardest to read. Lines 168-220.
Response: Thanks for the reviewer’s valuable comment. According to the suggestion of reviewer, we have made corrections according to this section. The amount of AgNWs increased from 1 ml to 6 ml, and a series of conductive electrodes are prepared and evaluated. With the increase of AgNWs, the transmittance of AgNW/PDMS and AgNW/PT-PDMS electrodes decreased (Figure 2b). When the load of AgNWs is 6ml, the transmittance of AgNW/PDMS electrode at 550nm is 37%, while that of AgNW/PT-PDMS electrode is 63%. The reason may be that AgNWs are different in the spreading region of AgNW/PDMS and AgNW/PT-PDMS electrodes. The whole surface of the AgNW/PDMS electrode is coated with nanowires, while in the AgNW/PT-PDMS electrode, the nanowires are only spread in the grooves. The area of nanowires covered by AgNW/PDMS electrodes is larger than that of AgNW/PT-PDMS electrodes, which leads to a serious decrease in transmittance. The conductivity of the electrode increases with the increase of AgNWs, as shown in Figure 2c. When the amount of AgNWs is 1ml, the conductivity of AgNW/PT-PDMS is better than that of AgNW/PDMS electrode, which may be due to the fact that there are more nanowires in the unit groove structure than on the PDMS surface. In order to balance the transmittance and conductivity of the electrode, we chose the amount of AgNWs solution of 4ml to prepare the electrode.
- How was the WO3 incorporated into the AgNW/PT-PDMS structure? Section beginning in Line226.
Response: Thanks for the reviewer’s valuable comment. According to the suggestion of reviewer, we have added a description of the selection. “Using electron beam evaporation technology, the active material WO3 was evaporated onto an AgNW/PT-PDMS substrate (thickness 300 nm, pressure 5×10-4 Pa) to obtain WO3/AgNW/PT-PDMS electrode. Meanwhile, WO3/AgNW/PDMS electrode were prepared for comparison.”
- Not mentioned are future work and optimization of other parameters to improve structureintegrity.
Response: Thanks for the reviewer’s valuable comment. According to the suggestion of reviewer, we have added a description of the selection. In future work, we consider using stainless steel templates with different diameters to adjust the micropore diameter to further optimize the performance of the electrode.
- In my opinion, there are too few citations (17 in total). Perhaps including growth of market orinterests in this area of wearable devices with statistics or percentages would more robustly showthe demand for this type of research.
Response: Thanks for the reviewer’s valuable comment. According to the suggestion of reviewer, we have added some research. Please see the yellow marking in Refferences in the revised manuscript.

Reviewer 3 Report
This paper demonstrated the fabrication of a stretchable transparent electrode using an Ag nanowire embedded inside the pattered PDMS for stretchable electrochromic cell application. There are merits in using the microgroove patterned PDMS in enhancing optical transmittance, electrical conductivity, and reliability. However, the novelty of this work is not highlighted by comparing/referencing similar stretchable works. Further, it needs a few modifications and additional inputs as given below prior to the acceptance.
1) Authors should provide state of the art table comparing a few of the stretchable devices (such as https://doi.org/10.1039/C5NR00313J, https://doi.org/10.1002/admt.202100804, https://doi.org/10.1007/s12274-022-4088-x, https://doi.org/10.1017/9781108882330 and so on) and highlight the advantage in comparison with previous works.
2) Rephrase the sentence “In the process of bending and stretching, the mechanical stability of the electrode is improved for the AgNWs are fixed in the concave and do not easy to slip and peel off.” From page 3.
3) The picture (digital photographic image) of the stainless steel film and the PT-PDMS will be helpful to understand the dimension of the pattern and their replication in the PDMS. Current Figure 1 alone is not sufficient enough to understand the roughness pattern.
4) The scale bar in Figure 1 is not correct (a & b, c & d got interchanged).
5) The schematic diagrams are needed to demonstrate how Ag nanowires are embedded inside PDMS using rod coating, how the embedded wire looks like, and the representation of wires while the peel test.
6) Similarly, the author should provide the electrochromic cell structure and the digital photographic image of the cell under bleaching and coloration states.
Minor English correction is needed for example "In the process of bending and stretching, the mechanical stability of the electrode is improved for the AgNWs are fixed in the concave and do not easy to slip and peel off."
Author Response
Our responses to Reviewers’ comments are listed as follows:
Reply to Reviewer 3’s Comments:
Reviewer #3: This paper demonstrated the fabrication of a stretchable transparent electrode using an Ag nanowire embedded inside the pattered PDMS for stretchable electrochromic cell application. There are merits in using the microgroove patterned PDMS in enhancing optical transmittance, electrical conductivity, and reliability. However, the novelty of this work is not highlighted by comparing/referencing similar stretchable works. Further, it needs a few modifications and additional inputs as given below prior to the acceptance.
- Authors should provide state of the art table comparing a few of the stretchable devices (such as https://doi.org/10.1039/C5NR00313J, https://doi.org/10.1002/admt.202100804, https://doi.org/10.1007/s12274-022-4088-x,https://doi.org/10.1017/9781108882330 and so on) and highlight the advantage in comparison with previous works.
Response: Thanks for the reviewer’s valuable comment. According to the suggestion of the reviewer, we have added the comparison of AgNW/PT-PDMS with other commercial materials. Please see the yellow marking in Table S1 in the revised supplementary.
Table S1 Optoelectronic properties and mechanical flexibility of Various Reported FTEs in comparison with AgNW/PT-PDMS
Samples |
Transmittance (%)/Sheet resistance (Ω/sq) |
Bending cycles |
Peeling off cycles |
Ref. |
AgNW/PEDOT:PSS/PET |
89.2/9.4 |
2000 |
30 |
1 |
MXene/AgNW-PVA |
52.3/18.3 |
1000 |
- |
2 |
AgNW-PVA |
87.5/63 |
250 |
- |
3 |
ZnO-PDMS |
- |
1000 |
- |
4 |
AgNW/PDMS |
75/20 |
1000 |
- |
5 |
AgNW/PET |
95/10 |
1000 |
- |
6 |
AgNW/PDMS |
-/0.5 |
1000 |
- |
7 |
AgNW@TiO2-PI |
95/43.2 |
5000 |
100 |
8 |
AgNW/PT-PDMS |
74/6 |
10000 |
500 |
Our work |
- Rephrase the sentence “In the process of bending and stretching, the mechanical stability of the electrode is improved for the AgNWs are fixed in the concave and do not easy to slip and peel off." From page 7.
Response: Thanks for reviewer’s valuable comment. We have made correction according to the reviewer's comments. The statements of "In the process of bending and stretching, the mechanical stability of the electrode is improved for the AgNWs are fixed in the concave and do not easy to slip and peel off." were corrected as " In the process of bending and stretching, it is not easy to slip and peel off due to the fixation of AgNW in the micropore groove on the surface of PDMS, which improves the mechanical stability of the electrode." Please see the yellow marking in the page 7 of the Results and Discussion in the revised manuscript.
- 3. The picture (diqital photographic image) of the stainless steel film and the PT-PDMS wil be helpful to understand the dimension of the pattern and their replication in the PDMS. Current Figure 1 alone is not sufficient enough to understand the roughness pattern.
Response: Thanks for reviewer’s valuable comment. According to the suggestion of the reviewer, we have added the diqital photographic image of the stainless steel film and the PT-PDMS. In order to better understand the preparation process of PT-PDMS substrate, the physical pictures were displayed. Figure 1a shows a photographic image of the stainless steel film with dense pore diameter of about 60 μm. Figure 1b shows physical photos of PT-PDMS and PDMS substrates. Compared with the smooth PDMS surface, the PT-PDMS surface has obvious roughness. However, the surface roughness of PT-PDMS has little effect on the transmittance of the film (as shown in Figure 1c, the transmittance at wavelength 550 nm is ~ 90%). Please see the yellow marking in Figure 2 in the revised manuscript.
Figure 1. Physical pictures: Stainless steel film (e); PDMS and PT-PDMS (f); PT-PDMS (g).
- 4. The scale bar in Figure 1 is not correct (a & b, c & d got interchanged).
Response: Thanks for reviewer’s valuable comment. We are very sorry for our incorrect writing about the scale bars in Figure 1 and have made correction. Please see the Figure 1 in the revised manuscript.
Figure 1. SEM images: PT-PDMS (a, b at different magnifications); AgNW/PT-PDMS (c, d at different magnifications).
- 5. The schematic diagrams are needed to demonstrate how Ag nanowires are embedded inside PDMS using rod coating, how the embedded wire looks like, and the representation of wires while the peel test.
Response: Thanks for reviewer’s valuable comment. According to the opinions of the reviewers, we have added a flow chart and a schematic diagram of the peel test. The scheme 2 shows how silver nanowires are embedded inside PDMS using rod coating and the state of AgNWs during peel testing. Firstly, the PT-PDMS with micro-porous structure is shown. The AgNWs solution was dripped onto the surface of PT-PDMS, and the solution was evenly coated with Meyer rod. AgNWs is pushed into micropore grooves by Meyer rods and connected to each other to form a network. After repeated adhesion with 3M adhesive tape, the AgNW network structure will not be destroyed because it is fixed in the micropore slot on the surface of PDMS.
Scheme 2 Schematic illustration for AgNWs are embedded inside PDMS process.
- 6. Similarly, the author should provide the electrochromic cell structure and the digital photographic image of the cell under bleaching and coloration states
Response: Thanks for reviewer’s valuable comment. According to the suggestion of the reviewer, we have added the electrochromic cell structure and the digital photographic image of the cell under bleaching and coloration states. The WO3/AgNW/PT-PDMS electrode exhibited switching stability with coloring and bleaching states. The colored state of the electrode is uniform as shown in Figure 4b, and the coloring and bleaching states is reversible under different voltages.
Figure 4. Transmittance spectra for the bleaching and coloration states (a); Photograph for the bleaching and coloration states (b); Electrochromic switching of WO3/AgNW/PDMS (c) and WO3/AgNW/PT-PDMS electrodes (d).
References
(1) Li, X.; Yu, S.; Zhao, L.; Wu, M.; Dong, H. Hybrid PEDOT:PSS to obtain high-performance Ag NW-based flexible transparent electrodes for transparent heaters. Journal of Materials Science: Materials in Electronics 2020, 31 (10), 8106-8115, DOI: 10.1007/s10854-020-03351-5.
(2) Zhou, B.; Su, M.; Yang, D.; Han, G.; Shen, C. Flexible MXene/Silver Nanowire-Based Transparent Conductive Film with Electromagnetic Interference Shielding and Electro-Photo-Thermal Performance. ACS Applied Materials & Interfaces 2020, XXXX (XXX).
(3) Zeng, X. Y.; Zhang, Q. K.; Yu, R. M.; Lu, C. Z. A new transparent conductor: silver nanowire film buried at the surface of a transparent polymer. Advanced Materials 2010, 22 (40), 4484-4488.
(4) Kumaresan, Y.; Min, G.; Dahiya, A. S.; Ejaz, A.; Shakthivel, D.; Dahiya, R. Kirigami and Mogul-Patterned Ultra-Stretchable High-Performance ZnO Nanowires-Based Photodetector. Advanced Materials Technologies.
(5) Liu, H. S.; Pan, B. C.; Liou, G. S. Highly transparent AgNW/PDMS stretchable electrodes for elastomeric electrochromic devices. Nanoscale 2017, 9.
(6) Highly Efficient and Bendable Organic Solar Cells with Solution-Processed Silver Nanowire Electrodes. Advanced Functional Materials 2014, 23 (34), 4272-4272.
(7) Lin, Y.; Li, Q.; Ding, C.; Wang, J.; Yuan, W.; Liu, Z.; Su, W.; Cui, Z. High-resolution and large-size stretchable electrodes based on patterned silver nanowires composites. Nano Research 2022, 15 (5), 4590-4598, DOI: 10.1007/s12274-022-4088-x.
(8) Huang, Y.; Tian, Y.; Hang, C.; Liu, Y.; Wang, S.; Qi, M.; Zhang, H.; Peng, Q. TiO2-Coated Core-Shell Ag Nanowire Networks for Robust and Washable Flexible Transparent Electrodes. ACS Applied Nano Materials 2019.

Reviewer 4 Report
Overall an interesting approach and results. Prior to publication:
The “self-made stretching machine” needs further explanation/description.
Were replicates characterized? None of the data shows error bars or uncertainty. This is a must-have for these materials.
seems fine
Author Response
Reply to Reviewer 4’s Comments:
Reviewer #4: Overall an interesting approach and results. Prior to publication:
- The "self-made stretching machine” needs further explanation/description.
Response: Thanks for the reviewer’s valuable comment. According to the suggestion of the reviewer, we have added the further introduction and description of the self-made stretching machine. The specific operation step of the self-made stretching machine is to bend or stretch the conductive electrode for a certain number of cycles, and test the resistance change in the testing process to get the mechanical stability of the electrode. Please see the yellow marking in the page 7 of the Results and Discussion in the revised supplementary.
- Were replicates characterized? None of the data shows error bars or uncertainty. This is a must-have for these materials.
Response: Thanks for the reviewer’s valuable comment. According to the suggestion of reviewer, we have added an error bar in the figure 3. Please see the yellow marking in the revised manuscript.
Figure 2. Transmittance spectra of PDMS and PT-PDMS Sheet resistance (c).

Round 2
Reviewer 4 Report
The authors have added error bars to Figure 2c, which is good. However the replicability of the results in terms of were replicates performed and error bars should also be added to at least the Figure 3 results and be described in the methods section.
Author Response
Reply to Reviewer 4’s Comments:
Reviewer #4: The authors have added error bars to Figure 2c, which is good. However the replicability of the results in terms of were replicates performed and error bars should also be added to at least the Figure 3 results and be described in the methods section.:
Response: Thanks for the reviewer’s valuable comment. According to the suggestion of the reviewer, we have added an error bar in Figure 3 and described it in the methods section. Please see the yellow marking on page 7 of the Experimental and Section.
Figure 3. Relative resistance changes under bending tests with 10,000 bending cycles (a); bending tests with 10,000 bending cycles (b); different tensile strain from 10% to 80% (c); taping tests with 500 cycles (d) of AgNW/PDMS, AgNW/PT-PDMS electrodes.
